# Comparing Visual-Only and Visual-Palpation Post-Mortem Lung Scoring Systems in Slaughtering Pigs

**DOI:** 10.3390/ani13152419

**Published:** 2023-07-26

**Authors:** Sergio Ghidini, Silvio De Luca, Elena Rinaldi, Emanuela Zanardi, Adriana Ianieri, Federica Guadagno, Giovanni Loris Alborali, Diana Meemken, Mauro Conter, Maria Olga Varrà

**Affiliations:** 1Department of Food and Drug, University of Parma, Via del Taglio 10, 43126 Parma, Italy; sergio.ghidini@unipr.it (S.G.); silvio.deluca@unipr.it (S.D.L.); elena.rinaldi@unipr.it (E.R.); emanuela.zanardi@unipr.it (E.Z.); adriana.ianieri@unipr.it (A.I.); mariaolga.varra@unipr.it (M.O.V.); 2Istituto Zooprofilattico Sperimentale della Lombardia e dell’Emilia-Romagna-Headquarters, Via A. Bianchi, 9, 25124 Brescia, Italy; federica.guadagno@izsler.it (F.G.); giovanni.alborali@izsler.it (G.L.A.); 3Department of Veterinary Medicine, Institute of Food Safety and Food Hygiene, Freie Universität Berlin, 14163 Berlin, Germany; diana.meemken@fu-berlin.de; 4Department of Veterinary Science, University of Parma, Via del Taglio 10, 43126 Parma, Italy

**Keywords:** pig, slaughter, lung, respiratory disease, scoring schemes, post-mortem inspection

## Abstract

**Simple Summary:**

Lung lesions, which are common findings in finisher pigs, are considered useful indicators of animal health and welfare at the slaughterhouse. Several methods for evaluating such lesions have been developed, and selecting the best system is critical for obtaining reliable and consistent data. In this view, the aim of this study was to compare two well-known scoring methods (Madec and Blaha methods) using data on lung lesions collected by two operators in an Italian high-throughput abattoir under routine slaughter conditions. Overall, there was a satisfactory level of agreement between the two methods, although a discrepancy in accurately recognizing healthy lungs and minor injuries between the two systems has been observed. According to our results, the Blaha method is a visual-only system applicable to very large abattoirs. It can be easily used to provide an overview of the respiratory health status of pigs, although it can yield a higher percentage of false negative results. On the other hand, the Madec method can provide more detailed results but would be more challenging to use for continuous monitoring in large abattoirs because it requires both visual and palpation assessments.

**Abstract:**

Respiratory diseases continue to pose significant challenges in pig production, and the assessment of lung lesions at the abattoir can provide valuable data for epidemiological investigations and disease surveillance. The evaluation of lung lesions at slaughter is a relatively simple, fast, and straightforward process but variations arising from different abattoirs, observers, and scoring methods can introduce uncertainty; moreover, the presence of multiple scoring systems complicates the comparisons of different studies, and currently, there are limited studies that compare these systems among each other. The objective of this study was to compare validated, simplified, and standardized schemes for assessing surface-related lung lesions in slaughtered pigs and analyze their reliability under field conditions. This study was conducted in a high-throughput abattoir in Italy, where two different scoring methods (Madec and Blaha) were benchmarked using 637 plucks. Statistical analysis revealed a good agreement between the two methods when severe or medium lesions were observed; however, their ability to accurately identify healthy lungs and minor injuries diverged significantly. These findings demonstrate that the Blaha method is more suitable for routine surveillance of swine respiratory diseases, whereas the Madec method can give more detailed and reliable results for the respiratory and welfare status of the animals at the farm level.

## 1. Introduction

Ante-mortem and post-mortem inspections play a vital role in ensuring food safety and compliance with European regulations on animal health and welfare [1,2]. In Europe, the regulations governing post-mortem inspection procedures in domestic pigs are outlined by the Commission Implementing Regulation (EU) 2019/627, which include visual inspection, additional palpation, incision of the carcass and offal, and laboratory tests as necessary. Additional post-mortem inspection procedures must be conducted when a potential risk to human health, animal health, or animal welfare may exist, as determined by the official veterinarian [1].

In recent years, there has been a notable shift in post-mortem inspections, moving away from routine palpation and incision towards visual-only post-mortem examinations. This change has allowed the slaughter industry to strike a balance between accurate official control and reduced risk of cross-contamination while remaining cost-effective [3,4,5].

Respiratory diseases in pigs, which arise from the interaction of infectious agents (viruses, mycoplasmas, and bacteria), host factors, and environmental conditions, pose significant challenges in the swine industry worldwide [6,7,8,9,10]. Swine respiratory diseases have had a considerable economic impact on intensive pig farming, resulting in higher morbidity and mortality rates, increased antimicrobial use, reduced feed conversion and growth rates, and increased carcass condemnation [6,9]. These circumstances highlight the need to improve the respiratory health of fattening pigs and emphasize the importance of research in enhancing pig farm management, strengthening biosecurity measures, and developing more effective vaccines [11].

Several factors, such as production system practices, ventilation, and flooring type can contribute to the development, transmission, and spreading of respiratory diseases on farms [10]. The occurrence of pneumonia in finisher pigs varies between 19% and 79% depending on factors such as location, production system, or the method used to assess lung lesions [12,13,14,15,16]. Within this context, broncho-pneumonic lesions, also known as Enzootic Pneumonia (EP)-like lesions, are the most prevalent ones. These EP-like lesions are characterized by purple to grey pulmonary consolidation areas and are mostly located in the cranioventral areas of the lungs [17].

Assessing lung lesions is crucial for identifying potential health issues that could affect meat quality and safety. Indeed, animals with respiratory conditions may have their meat downgraded or even condemned, based on the severity of the illness. Moreover, lung lesions may be indicative of welfare issues and may be included in the context of the abattoir-based measures because they can be visually evaluated by the assessors throughout slaughtering procedures [18]; however, the practice of palpating and cutting the lungs of pigs during slaughter poses several problems, including the risk of bacterial contamination of the carcasses and potential injuries to workers due to the use of sharp tools. On the contrary, visual assessment is well-aligned with the current approach of the visual-only inspection method used in European meat inspection [3,4,5].

The ideal scoring method for assessing lung lesions should be simple, fast, efficient, and standardizable. The methods currently used at slaughter generally meet these criteria but they are time-consuming and costly, hindering systematic lesion recording. Additionally, abattoir-related, and inter-observer variations may arise, highlighting the importance of standardizing all operative procedures [19,20,21].

Various scoring systems, including those proposed by Madec and Kobish [22], Christensen and colleagues [23], and Piffer and Brito [24], have been developed and used to assess respiratory diseases in pigs at slaughter [22,23,24]. While these methods may differ in their approach and have their respective strengths and weaknesses, they serve as valuable tools for estimating the severity of both clinical and subclinical infections [6,9].

In Italian slaughterhouses, the most widely used method for evaluating lung lesions is “Madec’s grid,” proposed by Madec and Derrien [21,25,26,27]. This approach still depends on manual examination of the lungs, which does not align with the latest provisions of the official control regulation, even if it offers numerous benefits; moreover, an accurate assessment of lung lesions using Madec’s grid requires experienced operators. Another scoring scheme suitable for its applicability in slaughterhouses is the Blaha system [28]. It involves the visual evaluation of lung lesions, based on their estimated extension. Both Madec’s grid and Blaha’s scoring system are considered reliable and effective for use in fast-paced pig abattoirs; however, a direct comparison of these two methods under standard slaughter conditions has not been conducted previously.

Therefore, the main objective of this study was to compare Madec’s grid and Blaha’s scoring system for assessing the extent of lung lesions in slaughtered pigs under routine working conditions on a slaughter line. 

## 2. Materials and Methods

### 2.1. Description of Madec’s and Blaha’s Lung Scoring Systems

In this study, Madec’s scoring method (MSM) and Blaha’s scoring method (BSM) were compared in commercial conditions to assess lung lesions in pigs at the abattoir level. 

The MSM was first described by Madec and Kobish in 1982 [22]. This method relies on the use of both visual inspection and palpation, which allows the operator to separate the pulmonary lobes and to palpate the lungs in case of doubtful situations (e.g., presence of artifacts such as blood aspiration). According to this method, each lobe is divided into quarters, and each lobe receives a value that ranges from 0 to 4 depending on the extension of the lesion, for a maximum value of 28 for each lung (Figure 1A). 

On the other hand, the BSM was developed by Blaha et al. in 1993 [28] and included in the GermanAVVLmH in 2009 (a system to assess lesions in pigs at the slaughterhouse implemented at the national level) [29]. This method is based on visual inspection only and it allows the classification of the assessed lungs into four classes ranging from 0 to 3, based on the estimated extension of lesions on the lung surface, as shown in Figure 1B. In detail, each of the cranial, medial, and accessory lobes accounts for a maximum of 10% of the lung surface, while each of the caudal lobes accounts for a maximum of 25% of the lung surface (in agreement with the estimation of lung surface described by Steinmann et al. 2014) [30] (Figure 1B). The overall lung score is calculated by adding up the percentages assigned to each lobe during the evaluation. A score of 0, 1, 2, or 3 is given to healthy lungs, lungs with lesions less than 10%, lesions between 11% and 30%, and lesions greater than 30%, respectively. 

An example of a swine lung lesion assessment using both MSM and BSM is provided in Figure 2. 

### 2.2. Training of the Observers

Two veterinarians (S.D.L. and E.R.), both authors of this study, were appointed to perform the lung assessment using the methods previously described. Both veterinarians had previous experience assessing lung lesions using the MSM, as this is a well-known method widely used in previous studies in Italian pig slaughterhouses [26,27]. Moreover, the MSM is the method included in the “ClassyFarm” project, an Italian integrated system that aims to categorize pig farms based on animal health-related data collected at national level on farms and at the slaughterhouses.

Prior to starting data collection, both veterinarians were trained in the use of the BSM by an expert in the field and author of this study (D.M.). The training consisted of two online sessions, focused in the first part on the description of the method and then on its adoption using pictures of lungs with different degrees of lesions previously collected during a different study. In this stage, specific attention was allocated to the differentiations between lung lesions and the presence of artifacts, such as blood aspiration and scalding water lungs, as these are reported as a frequent source of disagreement between observers during lung assessments at the slaughterhouse [21,30]. After the training, both operators conducted one-week full-day sessions at the same slaughterhouse where the data collection occurred to refine the adoption of the BSM and to identify any disagreements between the observers regarding the type and the extension of the lesions. In the event of discrepancies during the assessment, these were promptly discussed between the operators until reaching consensus about the score to assign. Finally, the inter-rater reliability between observers was calculated using a subset of ninety-two pictures of lungs previously collected during another study, with regard to both MSM and BSM. Cohen’s Kappa coefficient of more than 0.80 was reached and deemed sufficient to start the data collection.

### 2.3. Data Collection

This research was carried out in a high-throughput slaughterhouse in the Emilia-Romagna region (Northern Italy). The abattoir normally slaughters heavy pigs at a live weight of around 170 kg and at least 9 months of age, with a daily output of roughly 4500 heavy pigs intended for the Italian Protected Designation of Origin (PDO) product chain. The data collection was performed on swine lungs from different farms and batches during four sessions between July and August 2022 by the two trained veterinarians during the slaughter course. The first observer (S.D.L.) was appointed to assess the plucks using the MSM. A recording device placed in a pouch of the overall was used to record the findings, with the possibility to eventually palpate the organs in case of uncertainty. All the observed plucks were tagged in order to allow the second observer (E.R.) to locate and evaluate the same set of organs using the BSM. The BSM was performed exclusively through visual inspection, and all the results were contextually reported on paper. It should be noted that a gap of at least three plucks was left between the ones selected and tagged during each session to ensure both veterinarians had enough time for the assessments. The second observer was positioned about five meters from the first one, along the slaughter processing line and in proximity to the red offal room. Both the first and second observers’ locations were characterized by adequate lightning, and the distance between the two spots allowed an independent assessment of the identical set of organs without interference between operators. All the assessments were performed under routine slaughter conditions. At the end of each session, the findings of both observers were separately reported in an Excel file.

### 2.4. Statistical Analysis

The statistical analyses were performed in SPSS Statistics (IBM Corp. Released 2020, IBM SPSS Statistics for Windows, Version 28.0., Armonk, NY, USA).

The assessment results, which involved assigning final scores to the 637 plucks using the MSM and the BSM, were summarized using percentages and relative frequencies.

Three different statistical analyses were performed to test: (i) the consistency and reliability of the two different scoring systems; (ii) the differences between the groups tested with the two scoring systems; and (iii) the correlation between the results.

Specifically, the degree of accordance of the MSM and BSM was tested by calculating Cohen’s Kappa Measure of Agreement value (*p* < 0.05 significance level). In order to comply with the assumption of the test (equal number of categories for each response variable, i.e., for each scoring level) and achieve meaningful comparison and interpretation, the twenty-eight score points of the MSM were condensed and grouped into the four categories of the BSM, as reported in Table 1. The overall score distribution was considered, and proportionality between the methods was maintained with respect to the extension of lesions on the lung surface during the repartition of the MSM points across the BSM categories.

To identify statistically significant differences between the four groups of the BSM (categorical independent variable) on the ordinal dependent variable (MSM scores), the Kruskal–Wallis H test was applied. Bonferroni’s post hoc test (pairwise comparison of the groups) was performed to understand which of these groups differed from each other. The results were considered statistically significant if the *p* value was lower than 0.05.

Finally, the strength and direction of the correlation between the results obtained by using the two different scoring systems were tested using Spearman’s Rank Order Correlation analysis, with statistical significance set at a *p* value lower than 0.05.

## 3. Results

The distribution percentages of the prevalence of lung lesions assessed at slaughter on 637 plucks using the two cited scoring systems (MSM and BSM) are summarized in Figure 3. 

Based on the observations, a considerable number of the 637 assessed lungs received a score of 0, indicating a prevalence of 59% healthy lungs (373 out of the 637 assessed lungs) when the MSM was considered. By using the BSM, this prevalence was found to be 67% (428 out of the 637 assessed lungs). Hence, assuming the MSM as the gold standard, the BSM identified roughly 8% of lungs with “no lesions” (i.e., MSM-converted score and BSM score of 0), more than the MSM (Figure 3). In contrast, when the MSM was employed, a slightly greater number of lungs were classified as having minor, moderate, and severe injuries (i.e., MSM-converted score and BSM scores of 1, 2, and 3, respectively) compared to the BSM (Figure 3). Specifically, the MSM yielded scores of 1, 2, and 3 for 34%, 5%, and 3% of the lungs, respectively. On the other hand, when the BSM was employed, the corresponding percentages were 29%, 3%, and 1%.

Despite the above differences, the evaluation of the agreement between the two scoring methods resulted in a Kappa Measure of Agreement value of 0.698. This value suggests a generally satisfactory level of concordance in the classification of cases when the MSM is grouped into four categories [31]. 

The application of the Kruskal–Wallis test revealed a statistically significant difference in the continuous variable (MSM score) across the four BSM groups. The value of χ^2^ (3 df, *n* = 637) was 453.50, with a *p* value of 0.000. The fourth group of assessed lungs, which consisted of organs with severe injuries (MSM score = 3), exhibited a higher median score (*Md* = 10.00) compared to the other groups (Table 2).

The post hoc test showed which specific groups of the independent variable were significantly different from each other. As observed in Table 3, the *p* values for the comparison between groups 0–1 and 1–2 indicated the presence of a statistically significant difference in the score distribution based on the applied method (MSM or BSM) (*p* < 0.05); however, when comparing groups 2 and 3, this difference was not found to be statistically significant, and the results overlapped (*p* ≥ 0.05).

Finally, the relationship between the scoring methods was investigated using Spearman’s Rank Order Correlation analysis. There was a strong positive correlation between the two variables, with a coefficient of determination (*r*) of 0.81 (*p* < 0.001), with high MSM scores associated with high BSM scores.

## 4. Discussion

During the last decade, consumers and current policy have increasingly demanded improvements in animal welfare and health [32]. Overall, the slaughterhouse seems to be a convenient site for the observation of animal-based measures and for a comprehensive assessment of pig welfare, as well as the health management of the animals [18,33,34]. Meat inspection at the abattoir was primarily recognized as a tool to identify meat unfit for human consumption, while in recent years, experts in the field of animal welfare have explored its potential as a surveillance tool for animal health and welfare [33]. The strategic position of meat inspection at the slaughterhouse allows the assessment of an elevated number of pigs from different farms in a brief time and at a relatively lower cost compared to on-farm assessments, thus reducing the need for on-farm visiting [18].

Lung lesions are common findings in finisher pigs at the slaughterhouse [35]. The prevalence and severity of pneumonia at the abattoir appear to be good and useful indicators either to address control measures or to follow up on their efficiency at the farm level [26]. Indeed, by incorporating continuous feedback from slaughterhouses into herd management practices, there is a higher probability of improving farm performance gradually over time [36].

The detailed inspection of lung lesions at the slaughterhouse has the potential to offer a comprehensive overview of the respiratory health status of a farm [37,38,39]. It provides valuable insights into the variations in respiratory health among pigs raised in different sheds within the same farm, allows for the assessment of changes over time, and enables benchmarking against other farm groups or the national average. This makes slaughter data crucial in health programs and aids in raising farmers’ awareness of these lesions [40].

The evaluation of lung lesions at the slaughterhouse using scoring systems is generally considered a relatively simple and fast process; however, it is important to acknowledge that variation can arise from factors such as the specific abattoir, the observer conducting the assessment, and the chosen evaluation method [41]. Besides that, another challenge in this field is the existence of many scoring systems, which can make comparisons of findings across multiple studies difficult. Currently, there is limited research that specifically examines and compares the suitability and effectiveness of different scoring systems [16,18,21]; moreover, many of the existing studies have been conducted in laboratory settings rather than directly in real-world operating conditions. 

Indeed, throughout the years, several scoring methods have been developed to evaluate the severity of lung lobe lesions. These methods differ in terms of the time required for execution, the lesion extent on the lung surface, the level of accuracy, and ultimately, the resulting final score. These methods can be broadly classified into two-dimensional or three-dimensional scoring systems. In two-dimensional systems, the evaluation is primarily based on the spatial expansion of the lesion on the lung surface. In contrast, the three-dimensional scoring systems incorporate not only the lesion’s extent but also encompass the relative weight assigned to individual lung lobes, although their applicability for scoring lung lesions at the slaughterhouse may be limited [18]. 

In addition to the MSM and BSM, two-dimensional scoring methods have been proposed by Goodwin et al. (1969) [41], Hannan et al. (1982) [42], and Sibila et al. (2014) [43]. Goodwin and Hannan proposed point-based scoring systems, with Goodwin’s method assigning a maximum of 5 or 10 points to each lobe (depending on the lobe) for a total maximum score of 55, while Hannan’s method employed a schematic map of the lung to represent the areas with lesions, where the number of triangles affected by lesions in each lobe (varying based on the lobe size) is multiplied by five and divided by the total number of triangles in that lobe, resulting in a maximum achievable score of 35. In contrast, Sibila’s approach employs a scoring system based on percentages, similar to the BSM, utilizing image analysis of the lungs. This technique involves delineating the lesion area and the total area of the dorsal side of the lung in the image and utilizing digital analysis to calculate the proportion of the affected area relative to the entire area.

The analysis of the data collected in this present study revealed that when the MSM score is categorized into four classes, there is a satisfactory level of agreement between the classification of lung lesion cases using the MSM and the visual-only BSM; moreover, the study findings suggested that the BSM tends to identify a greater number of completely healthy lungs while simultaneously recognizing fewer lungs with minor, moderate, and severe lesions in comparison to the MSM (Figure 3). 

Therefore, this present study provides further confirmation that the two methods can lead to inter-rater disagreement, as previously reported by other researchers [44,45]. Nevertheless, it should be acknowledged that most scoring methods, such as the BSM, are based on visual and subjective estimation of the proportion of the affected lung surface and/or volume [18]. This situation can introduce the possibility of inconsistent evaluations of lesions, making scoring methods inevitably susceptible to errors. In particular, the situation described above could be attributed to the fact that the BSM is solely based on visual assessment. As a result, the BSM may fail to identify small lesions or lesions located in anatomical parts of the lungs that are not easily visible; consequently, this can lead to higher percentages of lungs being classified as completely healthy, as observed in Figure 3. Conversely, the MSM incorporates manual palpation and a more thorough assessment of the lungs, enabling the correct identification of those small lesions that may have been missed by the visual-only approach of the BSM. As a result, the prevalence of healthy lungs assessed using the MSM is lower, while the number of minor, moderate, and severe lesions is slightly higher (Figure 3). 

Therefore, based on the achieved results, it can be stated that the MSM can provide more detailed and reliable outcomes, especially for not- or minor-injured lungs. For this reason, the MSM proves to be a more valuable method for the evaluation of high-standard farm animals, presuming that they have a lower incidence of respiratory disease or when, for instance, the scope is to understand the performance of antimicrobial therapy or a vaccination strategy over time; however, the MSM is not easily applicable for routine and continuous monitoring of a large number of animals, as it requires manual palpation of the organs and should be applied by trained and specialized veterinarians. In contrast, the BSM is deemed more practical in this context and is well-suited for implementation in high-throughput plants, as it offers a comprehensive overview of the respiratory health status of the animals; moreover, although it may yield small percentages of false negative results, the BSM is better suited for use in association with computer vision systems, which primarily determine the percentage of the lungs‘ outer surface affected by lesions [46]. 

In conclusion, the limitations mentioned above, related to the slight disagreement among different lung scoring methods, could be effectively mitigated through the implementation of comprehensive veterinarian training programs that incorporate a standardized definition of the lesions [21]. This standardized definition would serve as a crucial reference point, ensuring consistency and reliability in the assessment process. By adopting this approach, it is anticipated that harmonization across different regions and abattoirs would be achieved, leading to a reduction in inter-rater variability and enhancing the overall accuracy of lung lesion evaluations. In the near future, it would be interesting to investigate and compare the data obtained from evaluations at the slaughterhouse with farm parameters affecting respiratory diseases.

## 5. Conclusions

From a public health perspective, the slaughterhouse plays a crucial role in ensuring meat safety and managing animal diseases. In particular, the surveillance of swine respiratory diseases is crucial for ensuring animal welfare and health. In this context, shifting the focus of control measures from farms to slaughterhouses can offer time and cost savings for both farmers and official authorities. The existing lung scoring systems, although effective, are unwieldy to use, requiring extensive data collection and time-consuming calculations for evaluation. Prior to assessing pneumonic lesions, the investigator must carefully select a scoring technique that is repeatable, suitable for the study’s objectives, and facilitates statistical analysis; therefore, choosing an evaluation system that is appropriate for the intended purpose is crucial to ensuring reliable and consistent data. 

The findings achieved in this present study indicate that the BSM for the evaluation of swine lung lesions can be considered a suitable method for routine surveillance of swine respiratory diseases. This method, which relies only on visual inspection, aligns well with current EU legislation and with the objective of minimizing cross-contamination of carcasses. Nevertheless, it was found that the visual-only approach of the BSM may lead to higher percentages of lungs being classified as healthy, potentially missing small or less-visible lesions. In contrast, the MSM, which incorporates manual palpation and a more thorough assessment, can better identify these small lesions, thus providing more detailed and reliable results regarding the respiratory and welfare status of the animals at the farm level.

These conclusions highlight the need to consider the specific goals and requirements of respiratory disease surveillance when selecting an appropriate scoring system.

## Figures and Tables

**Figure 1 animals-13-02419-f001:**
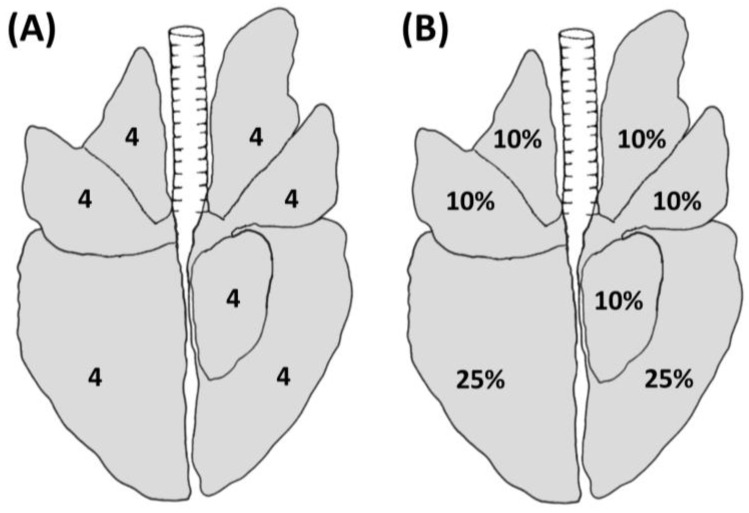
Schematic representation of lung lesions evaluated using the MSM (**A**) and BSM (**B**) depicting the maximum achievable score for each lobe.

**Figure 2 animals-13-02419-f002:**
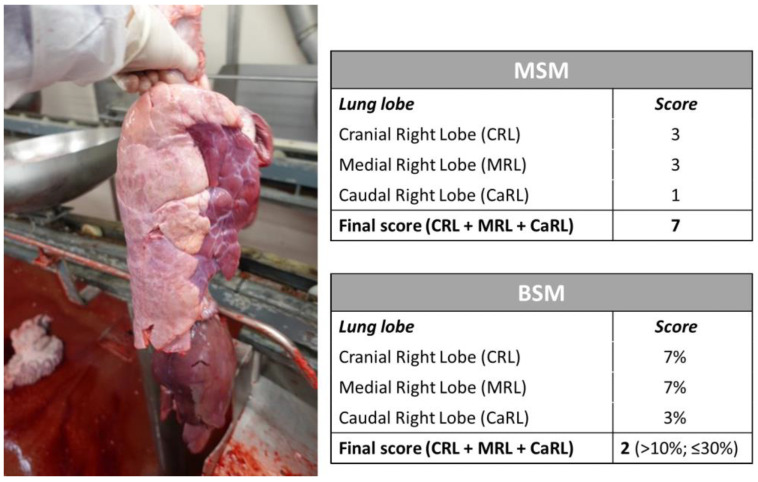
Example of lobe lesion assessment in a right lung using the MSM and BSM and resulting final scores.

**Figure 3 animals-13-02419-f003:**
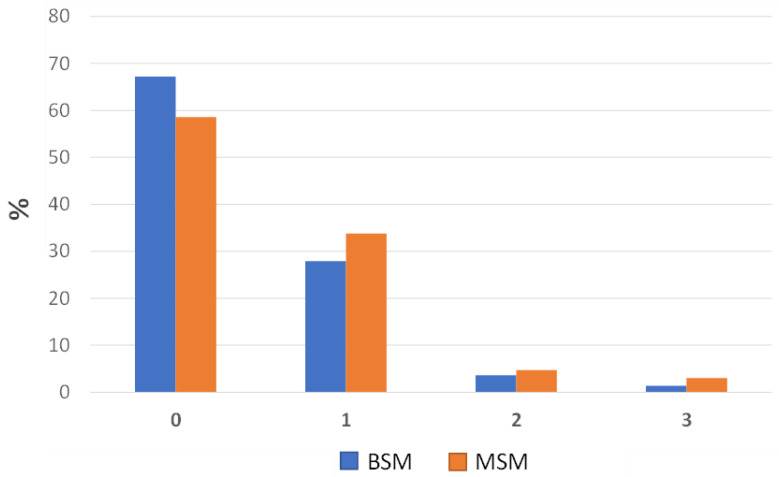
Bar charts showing the distribution percentages of the lung lesions across the four categories, as evaluated using the BSM and the MSM.

**Table 1 animals-13-02419-t001:** Grouping of MSM categories into the four BSM categories for statistical purposes.

MSM Score	BSM-Converted Score	Interpretation
0	0	No lesions
1–4	1	Minor injuries
5–8	2	Medium injuries
9–28	3	Severe injuries

**Table 2 animals-13-02419-t002:** Median score values of each MSM group (scores converted according to the BSM) used in the Kruskal–Wallis test followed by Bonferroni’s post hoc test.

MSM Score	Median Value	Number of Assessed Lungs
0	0.00	373
1	1.00	215
2	5.50	30
3	10.00	19
Total	---	637

**Table 3 animals-13-02419-t003:** Pairwise Comparisons obtained with Kruskal–Wallis test.

Sample	Test Statistic	Std. Error	Std. Test Statistic	Adj. Sig. ^a^
0–1	−273.665	14.613	−18.728	0.000
1–2	−106.913	37.027	−2.887	0.023
2–3	−20.538	64.830	−0.317	1.000

Each row tests the null hypothesis that the Sample 1 and Sample 2 distributions are the same. Asymptotic significances (2-sided tests) are displayed. Significance level set at 0.05. ^a^ Significance values have been adjusted using Bonferroni’s correction for multiple tests

## Data Availability

All data are available on request from authors.

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
