# Peer review of "Comparing Visual-Only and Visual-Palpation Post-Mortem Lung Scoring Systems in Slaughtering Pigs"

_animals, 2023, doi:10.3390/ani13152419_

Round 1

Reviewer 1 Report

The text of the manuscript describes a comparison of two post-mortem lung scoring methods in slaughtered pigs. Despite the fact that the topic of the work is interesting, it, unfortunately, has a number of shortcomings. The introduction is too long and mostly loosely related to the topic of the study. Most of the descriptions of the legal aspects of a post-mortem examination are redundant. Greater emphasis should be placed on the description of the methods and their use in practice and in scientific studies. The materials and methods section lacks a description of the scoring methods or references to them. In my opinion, the whole design is also the most important weakness of the study. The results section also needs improving. The discussion is poor and overloaded with unnecessary content.

Major comments

The introduction needs deep rewriting.

Lines: 48-89. In my opinion, this part of the introduction could be shortened because ration the aim of the study is weak.

Lines: 132-134. How “reliability of the proposed schemes under routine working conditions” was assessed in the study?

The materials and methods section lacks a description of the scoring methods or references to them. The specific procedure of lung assessment should be also described better.

The design of the study when the first observer assessed the lungs using one method and the second using another method is not reliable. There were not any repetitions. The study could be more valuable if the observer used both methods in many repetitions (different batches of swine carcasses).

The results are presented chaotically, which prevents their deeper analysis and assessment of the accuracy of the static tests.

Many editing mistakes need correction i.e. the title of Figure 1.

In the discussions is a lot of information that does not relate to the results and objectives of the experiment at all (Lines 222-276).

There is a lack of discussion about the influence of the observer on the results. It’s especially interesting taking into account the difference in assessing completely healthy lungs appears between observers.

Conclusions are not justified by the results of the study.

Reviewer 2 Report

The manuscript entitled by Ghidini et al. "Comparison of post-mortem lung scoring systems in slaughtering pigs" deals with the comparison between two scoring systems of evaluation lung lesions of swine in slaughterhouse.

The study applied two used (not new) scoring systems (MSM and BSM) to evaluate the lung lesions of slaughter pigs. Compare two scoring systems which depending on the vision and experience of 2 evaluator persons. Results of such evaluator are subjected to many human factors which are individual from the person to other.

There are some issues need to be considered:

manuscript need improve the language by English native speaker (I found some misspellings)

Introduction paragraph: is long

Material and methods: required more technician details about used scoring systems (MSM and BSM) and its application. Include the manuscript (in the material and methods) representative pictures for each lung lesion level (minor-1, medium-2 and severe injuries-3) if it is possible

Results paragraph: is short and brief

Discussion paragraph: in most of paragraph (lines 222-270) you reviewing the topic same as in the paragraph of introduction, whereas you discuss the results of your study briefly.

Conclusion paragraph: again you reviewing the topic same as in the paragraph of discussion and introduction and briefly focus in your conclusion according to the results of your study.  

Round 2

Reviewer 1 Report

The current version of the manuscript can be accepted for publication after doing another attentive text editing.

Author Response

The authors sincerely appreciate the reviewer’s thorough evaluation of the manuscript which improved the overall quality of the work. The authors have carefully considered the suggestion for another round of attentive text editing, addressing grammar, syntax, formatting, and punctuation errors throughout the entire document. The revised parts of the document have been highlighted using the track changes mode in MS Word.

Reviewer 2 Report

The revision was accepted 

Author Response

The authors express their gratitude to the reviewer for dedicating their time and expertise to evaluate the current study. They appreciate the reviewer's constructive feedback, which helped address multiple aspects of the paper and enhance its overall significance.